# Karnaugh-Veitch Maps as Minimal Formal Contract between Textual Requirements and Tests: A Use-Case Based Technical Analysis

Nils Henning Müllner 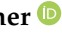

Institute of Transportation Systems, German Aerospace Center DLR, 38108 Braunschweig, Germany; nils.muellner@dlr.de

**Abstract:** Checking that requirements written in natural language hold for a formally implemented system is a complex task. Test steps are commonly implemented manually from the requirements. This process is inherently prone to mistakes, as test cases are complex and need to be analyzed sequentially to check which input/output combinations are tested (although tools allow for explicit tracing). Utilizing Karnaugh–Veitch maps as minimal formal contract between informal requirements and implemented test steps improves this process. KV-maps provide the requirements in a computer-editable way, as they correspond to Boolean formulas. KV-maps further allow to define which test steps are relevant. With both requirements and relevance specification at hand, test steps are automatically generated. The approach is applied on a real-world industrial use-case—a train control management system. Although being generally amenable to permutation testing, the selected use-case emphasizes the potential of the method. The method successfully demonstrates its benefits and may help to disclose flaws in the current manually implemented tests.

**Keywords:** IBM doors; KV-maps; karnaugh; veitch; permutation testing; TCMS

## 1. Introduction

A classic software life-cycle commonly comprises four steps: (i) requirements analysis, (ii) design, (iii) implementation, and (iv) test-case generation to check whether the final product satisfies the initial requirements, and may be enriched by agile approaches, dedicated deployment, or maintenance phases. Satisfying requirements is especially important for safety-critical systems. A flawed test-procedure can fail to detect flaws in the implementation (false negatives) or label a product useless, although it satisfies its requirements (false positives). While implementing test steps manually is inherently prone to mistakes, their detection is difficult, as the test result does not state if the test itself is correct (i.e., as intended).

Even if not potentially endangering lives, testing is also important in terms of liability. A defective product can ruin a company if it was not reasonably thoroughly tested. This paper aims at improving the trade-off between the costs of testing and the potential risk of having overlooked a potential safety hazard by showing how Karnaugh–Veitch-maps [1] can be exploited in testing, highlighting advantages and challenges.

KV-maps can simplify working with Boolean expressions. They were introduced by Veitch in 1952 [2] and refined by Karnaugh in 1953 [1] with an original focus on simplifying circuitry. The results of a Boolean formula are written as a truth table. The results are then transferred into a two dimensional grid such that each cell represents one input combination and its content represents the output. Optimal groups of 0 s and 1 s can be identified and the canonical form of the formula can be derived.

The research question is: How can the process of test step implementation be improved? This question is motivated by the state-of-the-art (i) being prone to flaws and (ii) being limited in the number of test steps in practice. One method to answer the question

is KV-maps. These can serve as minimal formal contract to avoid mistakes in the implementation of test steps and help to automatize the test step generation. We present a real industrial case study, a train control management system (TCMS), to demonstrate how KV-maps can help to overcome the limitations of manual implementation, which for the case study was still the state-of-the-art.

KV-maps offer a structural approach for linking test cases in their textual form to code (and even generate the code based on them). Alternatives from model-based testing are discussed in Section 2, yet no source could be found where KV-maps have been exploited in a similar way. Hence, this work does not focus on benchmarking or outperforming existing tools, but instead establishes a novel approach to overcome the limitations of the state-of-the-art, exemplarily demonstrated on a real industrial use-case, the TCMS.

Whenever requirements specify which combinations of (Boolean) signals are to be tested, a KV-map can be a handy tool to connect a requirement with its counterpart in the implementation. Furthermore, just linking that requirement to the cell in the KV-map allows for the code to be generated automatically, ensuring that all tests are conducted as expected. Exploiting KV-maps—here for once not for optimizing circuitry, but for permutation testing—hence has two benefits: Ensuring completeness and automatic test generation. To the best of our knowledge, KV-maps have not been utilized in this way until now and they have shown to be an invaluable asset for the presented use-case.

The TCMS, which serve as an example to demonstrate the method, is developed by Alstom Transportation AB (Formerly Bombardier Transportation when this work was conducted) in Västerås, Sweden. The company employs the tool IBM DOORS (Dynamic Object Oriented Requirements System) [3] for managing the requirements analysis, from which tests are manually implemented and executed on both a simulator and a physical system. For instance, the phrase "the alarm-light shall flash within one second if the sensor is activated" is implemented in C#, as shown in Listing 1.

**Listing 1.** Code Example, one Test-Step in the TCMS.

```
1  RTSIM.SIG_A.Force(true);
2  WaitForCondition(RTSIM.OUT_A, Is.Equal, 1, 1000);
```

The original signal names are exchanged for security reasons and the original textual requirements documents are closed source. The first line sets a sensor `SIG_A` to true and the second checks if the desired post-condition **RTSIM.OUT_A, Is.Equal, 1** holds within 1000 ms. Section 3 shows how manual implementation caused undetected mistakes in the test routine, which motivated to reason how such mistakes can be avoided. The formal approach in a nutshell is to

1.  write down the input vectors to be tested formally into a KV-map from the textual description, and
2.  either compare if manually implemented tests can be linked to all entries, or
3.  utilize that map for automatic implementation to guarantee full coverage.

Similarly, tools like IBM DOORS allow requirements to be linked to specific test steps. The advantage of KV-maps is that the test engineer only has to concern with few Boolean formulas and their relation to textual requirements instead of confusing test steps of a test case and their relation. Beyond that, KV-maps can be exploited to provide the simplified canonical form for automatic code generation. While KV-maps can serve as a graphical aid to visually confirm all required test steps are included, they serve here only as minimal formal contract and for test script generation. Being minimal here means that all possible test steps are represented in a (Boolean) matrix and all required test steps are labeled 1, while all others are labeled 0. The approach generally holds for non-Boolean domains also, and also for higher dimensions (as discussed in Section 6.2, Step 2). However, for the featured TCMS, this is not required. The contributions of this paper are:

1.  showing how KV-maps are established as a minimal formal contract between textual requirements analysis and implementation,
2.  introducing the TCMS and its test cases as an industrial case study, and,
3.  with KV-maps allowing to define which test steps are relevant, showing how KV-maps—specifying both requirements and relevance—can be further exploited for an automatic (and provably correct) test step generation.

The state of the art is discussed in Section 2 in the light of selected relevant literature. Section 3 methodologically differentiates the terms *bug*, *mistake*, and *fault* in the light of this paper, before Section 4 introduces the TCMS as system under test (SuT), along with the test suite and its corresponding KV-map in Section 5. Section 6 shows the custom-tailored script for automatic test step generation based on KV maps. Section 7 concludes this paper.

## 2. Related Work and State of the Art

Exploiting KV-maps for a structured approach to mistake-free test generation benefits from research from many domains. The goal of this section is not to provide a holistic survey over all available literature (which would be worth an article on its own), but to reflect the state-of-the-art in adjacent domains.

Why are the test cases implemented manually in the first place? What are the alternatives? The motivation for the manual approach is that each functionality is simply tested as a unit. An integration test covering cross-functionality is not demanded for the featured case study, and as each unit is sufficiently small (with six or 13 steps to implement for the three sensors featured here), more complex approaches are too expensive. This brings back the question for the ideal trade-off between safety and amount of testing from Section 1.

Model-based approaches seem like a nice possibility to maximize the testing coverage with automated test-step generation. For instance, Alrawashdeh (On the paper, the author's name seems to be misspelled as Alrawashed) et al. [4] provided an ATM system as SuT and discussed model-based testing, improved with genetic algorithms. For systems that are too large for being holistically tested, optimizing the test step selection is important. For the TCMS, on the other hand, KV-maps allowed for achieving full integration test coverage. The actual bottleneck in the script presented for the TCMS is rather the slow writing of the test steps into a file on a hard disk, not its generation in the RAM. We did not discover the limit of the test case generation with KV-maps. That (depending on the hardware provided) is yet to be found with larger challenges. For such challenges, Tomita et al. [5] propose a tool exploiting Monte Carlo test-case generation, implying the crucial question: Why write a new tool?

KV-maps define a minimal formal contract hinging text to code. To the best of our knowledge, they have not been established yet for challenges like the TCMS. Second, the tools considered:

*   Spec explorer (https://www.microsoft.com/en-us/research/project/model-based-testing-with-specexplorer/, last visited 30 May 2022) focuses on design conformity,
*   MaTeLo (https://matelo.all4tec.com/, last visited 30 May 2022) focuses test generation via graphical design,
*   Conformiq (https://www.conformiq.com/, last visited 30 May 2022) focuses on improving testing with AI, and
*   Smartesting (https://www.smartesting.com/?lang=en, last visited 30 May 2022) focuses on cognitive test automation,

tackle different challenges and do not exploit KV-maps. The initial motivation to utilize KV-maps was to have a minimal formal contract that is as clear and as short as can be. Instead of looking for a tool fitting the TCMS we looked for a method and then if there is a tool featuring that method. With a lack of a proper tool, we implemented our own tool which is tailored to the TCMS, whereas the method is generally applicable. Exploiting formal methods for model-based test generation, for instance discussed by Kalaee and Rafe [6], exemplarily shows how academic contributions are disseminated in industrial contexts. Similarly, KV-maps can be exploited.

The closer a deadline for release gets, the less testers commonly have to do. According to Spadini et al., providing "high-quality test code is [...] considered of secondary importance" [7]. They address this issue and provide a study containing interviews of twelve developers, analyzing more than 300,000 code reviews on a meta-level. Our paper addresses their third research question: "Which practices do reviewers follow for test files"? Exploiting KV-maps as described here provides a structured method for generating test files with a minimal formal contract as evidence.

A less formal notion for addressing the quality of testing are *smells* in software code. For instance, Garousi and Küçük [8] provide a literature survey for that domain, including a similar discussion about the differences of manual and automated testing. With KV-maps at hand, smells become redundant. An earlier study provided by van Deursen et al. [9] puts practical lessons learned to the smells of code and can be very helpful when KV-maps cannot be employed. Kitamura et al. [10] provide graphical notations for modeling tests in combinatorial testing tailored for tree-structured strategies. Although their scope is different, their approach can be extended to discuss permutation testing based on KV-maps in the future. Regarding combinatorial testing, Kuhn et al. [11] exemplarily discuss how exhaustive testing can be avoided. This topic is addressed in Section 4.1 by pointing out that classes of bisimilar test steps require only one representative step to be tested. Further model-based testing methods, to which our approach can be applied to, are discussed by Utting et al. [12]. The exploitation of a minimal formal contract between textual requirements and actual test implementation (or their automatic generation based on those contracts) is not part of this survey. Im et al. [13] show a tool-chain for a model-driven creation of test cases, which are automatically extracted from use cases that are specified in a domain-specific language (DSL, [14,15]).

Despite model-based testing, other areas are also important. Regarding high-level system specification, Ferrarotti et al. [16] propose a method for transferring Java source code into Gurevich's Abstract State Machines to describe the behaviour of the system without irrelevant details with the goal of improving current software engineering practices and testing. With state-transitions being redundant to the TCMS, we can focus on a different approach. Exemplary for correct-by-construction approaches, Benyagoub et al. [17] advocate for verifying conversation protocols of distributed systems. Although communication is vital for the TCMS, it is not its test subject. Zhuang [18] presents *PuPPy* as an extension to Python for helping programmers in declaring fields (class attributes) as signals, similar to the signals in the TCMS featured in this paper. Both explicit events and implicit signals can be regarded as a mechanism to propagate value changes of variables. PuPPY provides a lightweight push-pull model to use variables as signals in imperative programming focusing on object-oriented design. Naumchev et al. [19] present a tool-supported method, AutoReq, for programming with contracts. It produces verified requirements which are demonstrated on the use-case of a landing gear system. The question for a minimal formal contract is not discussed. Kos et al. [20] show that one can leverage on the domain-specific modelling languages by implementing assertions to it to build a testing framework with relatively little effort, similar to the utilization of KV-maps for generating test steps.

## 3. Mistake Propagation

A requirement analysis describes what has to be tested along with the expected outcomes. For the case of the TCMS, it is manually transferred to executable code, setting the signals and waiting for the expected outcome. This manual transfer is prone to mistakes, caused by regular typos (e.g., setting the wrong signal or the wrong value). They can be structural by misinterpreting the requirements and they can propagate. Since the setting of signals follows a certain pattern, exploiting model-based testing for autonomous code generation seems attractive.

The

1. SuT,
2. the test case qualifying the system against desired properties, and
3. the environment in which all are executed

determine the actual execution of the testing sequence. A bug in the SuT (1.) might cause undesired behavior, as well as the environment in which the system operates (3.) when it is not within the limits for which the system is specified. The middle item (2.) addresses undesired behavior when the specification of the behavior itself is flawed while both system and environment are behaving as desired.

Undesired behavior can occur in each of these and the effects might spread through the system or test-case. This section proposes the following terminology for distinguishing between them shown in Table 1. Since the term *bug* is coined in software engineering and *fault* in fault tolerance, we select the term *mistake* to distinguish undesired behavior in testing.

**Table 1.** Terminology distinguishing Bugs, Mistakes, and Faults.

|  | **SuT** | **Testing** | **Environment** |
|---|---|---|---|
| causation | bug | mistake | fault |
| nature | deterministic | | probabilistic |

Mistake here means unintentional human-induced cause for unintended behavior during testing. The term *programming bug* refers to bugs in the SuT, faults to sporadic influence from the environment, and mistakes are *bugs* in testing.

The term *propagation* in that relation was first defined for *fault propagation*, referring to the spreading of effects of faults through the components of a system (cf. [21]). The term *bug propagation*, coined in software engineering [22,23], addresses unintended but system-internal causes for deviation from the intended behavior spreading through the system. Similarly, mistakes can spread through a sequence of test steps. Analogously, *mistake propagation* addresses persistence, continuation, or consequences of mistakes in testing sequences. Therefore, here,

- bugs are in software,
- mistakes in the testing procedure,
- and faults caused by the environment (potentially affecting either).

Their propagation accounts for spreading their effects within the system or testing procedure.

**Example 1.** *Consider three Boolean signals A, B, and C and the following test procedure in Listing 2 in pseudo code.*

This procedure tests all eight combinations, traversing the state space as shown in Figure 1 with solid black arrows. Now consider a mistake typo in TestStep 2, line 4, assigning $B = 1$; instead of $A$. The result is the trace shown by red dotted arrows. The trace misses to test three states.

**Listing 2.** Test Steps Exemplifying Mistake Propagation.

```
1  %TestStep1
2  A = 0; B = 0; C = 0; test();
3  %TestStep2
4  A = 1; test();
5  %TestStep3
6  B = 1; test();
7  %TestStep4
8  C = 1; test();
9  %TestStep5
10 A = 0; test();
11 %TestStep6
12 B = 0; test();
13 %TestStep7
14 A = 1; test();
15 %TestStep8
16 A = 0; B = 1; C = 0; test();
```

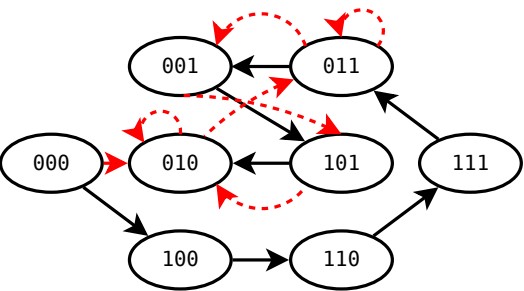

**Figure 1.** Teststepping through the State Space.

This example shows how a mistake can propagate through further test steps. Section 6.2 addresses this challenge in the light of the featured TCMS example.

## 4. The Train Control Management System and KV-Maps

The TCMS is a proprietary embedded system that is available as both a simulator (BT SoftTCMSController software, Version 1.1.5.0) and a real physical system. The simulation mimics the behavior of the original system, conservatively realistic wrt. timing properties. The three featured test cases are

1. smoke/fire sensor,
2. battery chargers, and
3. derailment sensors.

A signal is either a sensor signal or a validity signal of a sensor signal. It is a Boolean flag where true means *the good thing* [24], i.e., that no alarm shall be raised or that a signal is valid. The false value means *the bad thing*, i.e., that an alarm is to be raised or a signal is invalid. A set of sensors (e.g., two smoke/fire sensors) together with validity signals (one per sensor) are the *input* for one test case. For instance, two smoke/fire sensors with two validity signals make up the four inputs for the first test case *smoke/fire detectors*. The *output* signals are to be computed as described in the textual requirements analysis. For instance, consider that an alarm shall be raised if (i) at least one valid signal wants to raise the alarm, or (ii) if there is no valid signal.

Utilizing KV-maps for each individual test case shows how the mapping from input (i.e., position of the cell in the KV-map) to output (i.e., the value of the cell) is conducted and how this is helpful when checking the implementation. Once the KV-map is established,

we show how it can be further exploited for generating the test steps automatically. Beyond that, we discuss how combinatorial testing, i.e., here combining different test cases to common outputs, can be achieved, while still keeping the test case specific KV-maps separated. Section 4.1 shows, in general, how to derive a KV-map on the example.

### 4.1. From Textual Requirements to KV-Maps

Formalizing the smoke/fire sensors in this example requires four variables:

- `SIG_A` as signal from the first sensor with
- `SIG_AV` being its validity signal, and
- `SIG_B` as signal from the second sensor with
- `SIG_BV` being its validity signal.

The textual requirements state that an alarm shall fire if one valid alarm fires and there is at least one valid signal. This also means, in turn, that no alarm is sounded if only an invalid signal fires an alarm, and that an alarm is fired if both signals are valid but contradict each other. The formula

$$
\begin{aligned}
s \models \mathcal{P} : \neg((\neg \texttt{SIG\_A} \wedge \texttt{SIG\_AV}) \\
\vee (\neg \texttt{SIG\_B} \wedge \texttt{SIG\_BV})) \wedge \\
(\texttt{SIG\_AV} \vee \texttt{SIG\_BV})
\end{aligned}
\tag{1}
$$

reads: *The system state s satisfies (safety) predicate $\mathcal{P}$ if neither sensor raises a valid alarm, and there is at least one valid signal.* This formula can be written as Truth Table 2.

**Table 2.** Fire Sensor Truth Table.

|  |  | SIG_A SIG_AV | SIG_A ¬SIG_AV | ¬SIG_A SIG_A | ¬SIG_A ¬SIG_AV |
|---|---|---|---|---|---|
| SIG_B | SIG_BV | 1 | 1 | 0 | 1 |
| SIG_B | ¬SIG_BV | 1 | 0 | 0 | 0 |
| ¬SIG_B | SIG_BV | 0 | 0 | 0 | 0 |
| ¬SIG_B | ¬SIG_BV | 1 | 0 | 0 | 0 |

That thruth-table used as KV-map ( The online calculator https://www.mathematik.uni-marburg.de/~thormae/lectures/ti1/code/karnaughmap/, last visited 30 May 2022, generated all KV-maps in this paper) shown in Figure 2 derives the disjunctive normal form (DNF) shown in Equation (2):

$$
\begin{aligned}
s \models \mathcal{P} : ( \ \texttt{SIG\_A} \ \wedge \texttt{SIG\_AV} \wedge \neg \texttt{SIG\_BV}) \vee \\
(\neg \texttt{SIG\_AV} \wedge \texttt{SIG\_B} \ \wedge \ \texttt{SIG\_BV}) \vee \\
( \ \texttt{SIG\_A} \ \wedge \texttt{SIG\_AV} \wedge \ \texttt{SIG\_B} \ )
\end{aligned}
\tag{2}
$$

Notably, the graphical tool shown in Figure 2 is here only applied for a visual representation. For the formal approach, a coded representation suffices. One might think that adding KV-maps would add another source for mistakes. Such *loose ends* (unconnected values in the KV-map), which are either a textual requirement that is not connected or a missing implemented test step, are yet alerts, which is the reason to have this minimal formal contract in the first place. KV-maps here serve as a perfect detector verifying that all required test steps are included. Hence, adding a KV-map does not open a new vector for mistakes, but rather an additional check to avoid them.

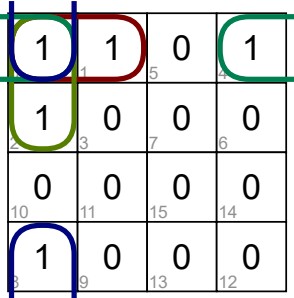

**Figure 2.** Fire Sensor KV-Maps.

A complete test would set the system to every single possible combination (there are 16) and check if the output is as expected (i.e., 1 when $s \models \mathcal{P}$ and 0 otherwise). For *certifying* the system, it is only required, that *one* representative combination for each requirement is tested. For instance, when one sensor is valid and true while the other is invalid, the test step in which the other sensor is valid and true with the first being invalid has not to be tested anymore. Both test steps are *bisimilar* (wrt. $\mathcal{P}$), and they belong to the same *equilvalence class*. *How* sets of steps qualify for *lumping* [25] is specified in the textual requirements analysis. The distinction between **test cases** (e.g., derailment sensors or smoke sensors) and **test steps** (i.e., a specific allocation of signals within a test case, or *system state*) is important. A test case (e.g., smoke/fire sensors) comprises several test steps (e.g., an allocation of values for the relevant signals). All test steps must succeed for a test case to pass. In case the alarm is activated when not intended to, the corresponding test step is labeled *false positive*, or *false negative* (i.e., an undetected fault) otherwise.

## 5. Testing

A common test step in the TCMS example has the form shown in Listing 3 in C#. The input signals are set (i.e., forced to either **true** or **false**) and the expected output condition has to hold within one second (here as 1000 ms).

**Listing 3.** An Exemplary Step.

```
1  RTSIM.SIG_A.Force(true);
2  RTSIM.SIG_AV.Force(true);
3  RTSIM.SIG_B.Force(true);
4  RTSIM.SIG_BV.Force(true);
5  WaitForCondition(RTSIM.OUT_A, Is.Equal, 1, 1000);
6  WaitForCondition(RTSIM.OUT_B, Is.Equal, 1, 1000);
7  WaitForCondition(RTSIM.OUT_C, Is.Equal, 1, 1000);
```

Notably, the requirements analysis states that all three outputs are expected to comply within the same one second. However, the script permits each signal in sequence up to one additional second. This mistake is ignored here, as technically all outputs are set simultaneously and cannot take advantage of the additional time to comply.

The following three sections explain the three test cases in detail. The TCMS example is well suited to show

1. why manual test case implementation is still the state of the art,
2. how mistakes are possible and not unlikely in such cases,
3. the shortcomings of direct implementation from requirements,
4. how KV-maps for automatic test step generation are not only an improvement over manually implemented test cases, but also unlock the potential to tackle larger test campaigns that either include more test cases or demand full integration testing,

thus improving the state of the art. Featuring three test cases, we introduce input variables with a leading

- `SIG1_` for test case 1, smoke/fire sensor,
- `SIG2_` for test case 2, battery chargers, and
- `SIG3_` for test case 3, derailment sensors.

The first signal of a test case is labelled `SIG*_A` and its validity signal (In the TCMS, every signal has a validity signal) `SIG*_AV`, the second `SIG*_B` and `SIG*_BV`, and so on. Each test case has one individual output labelled `OUT*`. All test cases also share three outputs per design specification labelled `OUTSHARE1`, `OUTSHARE2`, and `OUTSHARE3`. Paragraph Step 4 in Section 6.2 explains how they are computed during integration testing (i.e., when multiple test cases are tested at once).

### 5.1. Smoke/Fire Sensor

The smoke and fire sensors have two data signals and two validity signals labeled `SIG1_A`, `SIG1_AV`, `SIG1_B`, and `SIG1_BV`. Each signal can be set to **true** or **false**, which results in $2^4 = 16$ possible combinations for the first test case. The alarm shall become active within one second if (i) one valid sensor raises the alarm, or (ii) no sensor is valid. The alarm *output* is split into fours signals. All shared outputs are for redundancy purposes expected to be equal by design at all times.

Not all possible 16 test cases are relevant. It suffices to test one representative for each equivalence class (cf. Section 4.1). Table 3 shows the relevant test steps and their outputs.

**Table 3.** Test Case 1—Smoke and Fire.

| | **Inputs** | | | | **Outputs** | | | |
|---|---|---|---|---|---|---|---|---|
| **Step** | SIG1_A | SIG1_AV | SIG1_B | SIG1_BV | OUT1 | OUTSHARE1 | OUTSHARE2 | OUTSHARE3 |
| 1 | 1 | 1 | 1 | 1 | 1 | 1 | 1 | 1 |
| 2 | 1 | 0 | 1 | 1 | 1 | 1 | 1 | 1 |
| 3 | 1 | 0 | 0 | 0 | 0 | 0 | 0 | 0 |
| 4 | 1 | 1 | 0 | 0 | 1 | 1 | 1 | 1 |
| 5 | 1 | 1 | 0 | 1 | 0 | 0 | 0 | 0 |
| 6 | 0 | 1 | 0 | 1 | 0 | 0 | 0 | 0 |

### 5.2. Battery Chargers

The battery chargers comprise eight input signals from the sensors: Four original signals and four validity signals, labeled `SIG2_A`, `SIG2_B`, `SIG2_C`, `SIG2_D`, and `SIG2_AV`, `SIG2_BV`, `SIG2_CV`, `SIG2_DV`. There are $2^8 = 256$ possible test steps of which only 13 are deemed relevant, shown in Table 4. Signals `SIG2_B` and `SIG2_C` (and their validity signals respectively) are identical. Although they can be set to different values in theory, such a case never occurs as it is physically impossible. This feature is documented in the internal documentation. Hence, they are treated as one signal, indicated by the brackets in Table 4. This reduces the number to practically six signals, or $2^6 = 64$ possible combinations.

**Table 4.** Test Case 2—Battery Chargers.

| Step | SIG2_A | ⟨SIG2_B | SIG2_C⟩ | SIG2_D | SIG2_AV | ⟨SIG2_BV | SIG2_CV⟩ | SIG2_DV | OUT2 | OUTSHARE1 | OUTSHARE2 | OUTSHARE3 |
|---|---|---|---|---|---|---|---|---|---|---|---|---|
| 1 | 1 | 1 | 1 | 1 | 1 | 1 | 1 | 1 | 1 | 1 | 1 | 1 |
| 2 | 0 | 1 | 1 | 1 | 1 | 1 | 1 | 1 | 1 | 1 | 1 | 1 |
| 3 | 0 | 0 | 0 | 1 | 1 | 1 | 1 | 1 | 0 | 0 | 0 | 0 |
| 4 | 1 | 0 | 0 | 1 | 1 | 1 | 1 | 1 | 1 | 1 | 1 | 1 |
| 5 | 0 | 1 | 1 | 0 | 1 | 1 | 1 | 1 | 0 | 0 | 0 | 0 |
| 6 | 1 | 1 | 1 | 0 | 1 | 1 | 1 | 1 | 1 | 1 | 1 | 1 |
| 7 | 1 | 0 | 0 | 0 | 1 | 1 | 1 | 1 | 0 | 0 | 0 | 0 |
| 8 | 1 | 1 | 1 | 1 | 0 | 1 | 1 | 1 | 1 | 1 | 1 | 1 |
| 9 | 1 | 1 | 1 | 1 | 0 | 0 | 0 | 1 | 0 | 0 | 0 | 0 |
| 10 | 1 | 1 | 1 | 1 | 1 | 0 | 0 | 1 | 1 | 1 | 1 | 1 |
| 11 | 1 | 1 | 1 | 1 | 0 | 1 | 1 | 0 | 0 | 0 | 0 | 0 |
| 12 | 1 | 1 | 1 | 1 | 1 | 1 | 1 | 0 | 1 | 1 | 1 | 1 |
| 13 | 1 | 1 | 1 | 1 | 1 | 0 | 0 | 0 | 0 | 0 | 0 | 0 |

The next simplification comes from the requirements analysis: The output shall be 1 when there are at least two valid true input signals (treating `SIG2_B` and `SIG2_C` as one).

Since battery chargers are less critical than smoke/fire sensors, one valid alarm does not necessarily lead to an alarm. Instead, majority consensus applies. The first six steps in Table 4 (after the initialization step) iterate through the actual signals being set to trueor falsewhile all validity signals are true. Then, the exact same set of combinations is executed on the validity signals, while the actual signals are true. Thus, actual signals and validity signals can be treated as equal partitions, with each partition requiring the same six steps (plus one initialization step where all signals are true). The output signals comprise one individual signal `OUT2` and the three shared signals. Table 4 shows all relevant input combinations with their expected output.

*5.3. Derailment Sensors*

The derailment sensors have four inputs and four outputs. This test case shares the same relevant input combinations with the first test case as shown in Table 5. Notably, the expected output differs due to different safety constraints. In case of smoke/fire, one valid alarm or no valid sensor trigger the alarm. In case of derailment, even one inactive sensor (i.e., validation channel set to **false**) triggers the alarm. Only when all signals are **true** and valid is the (safety) predicate satisfied.

**Table 5.** Test Case 3—Derailment Sensor.

| Step | SIG3_A | SIG3_AV | SIG3_B | SIG3_BV | OUT3 | OUTSHARE1 | OUTSHARE2 | OUTSHARE3 |
| --- | --- | --- | --- | --- | --- | --- | --- | --- |
| 1 | 1 | 1 | 1 | 1 | 1 | 1 | 1 | 1 |
| 2 | 1 | 0 | 1 | 1 | 0 | 0 | 0 | 0 |
| 3 | 1 | 0 | 0 | 0 | 0 | 0 | 0 | 0 |
| 4 | 1 | 1 | 0 | 0 | 0 | 0 | 0 | 0 |
| 5 | 1 | 1 | 0 | 1 | 0 | 0 | 0 | 0 |
| 6 | 0 | 1 | 0 | 1 | 0 | 0 | 0 | 0 |

**6. Test Generator**

The starting point for understanding the test cases includes (i) the requirements analysis and (ii) the manually implemented test scripts in C# derived from the requirements. The available scripts initialize the system by setting all relevant input signals to **true** (cf. Listing 3). Afterwards, the system is set to different input configurations—by changing those values that differ from the previous configuration—and it is tested, if the system provides the expected output within 1 s. After executing all steps, the system is brought to a post condition in which it can shut down. Failed steps where the output is not as expected do not stop the test sequence. For test cases 1–3, the order in which the steps are executed (i.e., their sequence) is arbitrary. Safety is only state based, not trace-based. This means that safety is defined here momentarily over the one state and not a sequence of states over time.

*6.1. Manual Test Scripts*

The manually written test scripts set only those signals that change with regards to the previous system state (similar to the script discussed in Listing 2). For instance, the implementation of the first two steps in test case 1 from Table 3 read as shown in Listing 4 in C#:

Test step 2 in lines 8–11 only sets signal `SIG_AV`. This accelerates test execution as redundant signal-setting is avoided. Setting one signal takes about 20 ms in the simulator. Studying the manually implemented test cases revealed a mistake in a script, where the wrong signal was updated. Since only signals that change with regards to the previous step are updated, such mistakes can manifest until they are overwritten by another update as discussed in Figure 1. Therefore, even if the sequence of steps is not relevant for the test itself, it is relevant for mistake propagation. Undetected mistakes (false negatives or false positives) cannot occur in the test routine with KV-maps. Notably, the mistaken test step from the manual implementation here failed neither the unintended configuration nor the

correct one. The mistake was also corrected by chance directly in the following step by overwriting the affected signal before it could propagate through the following steps.

**Listing 4.** Two Sequential Test Steps Setting only Changing Signals.

```
1   /** Step 1 **/
2   RTSIM.SIG_A.Force(true);
3   RTSIM.SIG_AV.Force(true);
4   RTSIM.SIG_B.Force(true);
5   RTSIM.SIG_BV.Force(true);
6   WaitForCondition(RTSIM.OUT1, Is.Equal, 1, 1000);
7   /** analogously for OUTSHAREs**/
8   /** Step 2 **/
9   RTSIM.SIG_AV.Force(false);
10  WaitForCondition(RTSIM.OUT1, Is.Equal, 1, 1000);
11  /** analogously for OUTSHAREs**/
```

However, the mistake raised the question as to whether or not all signals should be set each time. When implementing manually, changing all signals each step takes longer and means higher chances of mistakes (based on human nature). Setting all signals each time is a bad option. For automatic implementation on the other hand, such mistakes do not occur. Setting all signals every step takes longer (raising complexity linear in the number of signals). A benefit of setting all signals is interchangeable steps. Since the system is to be tested, it is expected to violate the requirements. Testers want to re-test flagged configurations. A failed test step including its full configuration is a clear benefit over the reconstruction of a configuration over the trace of past steps. Additionally, fully set configurations allow for concurrent testing to distribute a stack of steps onto multiple systems or simulators.

The manually implemented test cases execute sequentially. Although the test cases share output signals, the requirement analysis does not demand testing test cases together (i.e., a shared output is only **true** when all its inputs are **true**). Without a requirement for integration testing, test cases can be dealt with one by one, and alone they do not face state space explosion. A state space explosion occurs when the number of states is exponential in the number of processes or signals constituting the states. In case integration testing would be required, $6 \times 13 \times 6 = 468$ test steps would have to be implemented instead of $6 + 13 + 6 = 25$ (i.e., exponential vs. linear complexity) for the three featured test cases.

The TCMS test suite is a real industrial example, suitable to investigate how testing a larger system can be accomplished in the future. Assuming linear complexity at this point just because the requirements analysis does not explicitly require integration tests is dangerous, especially when test cases share common outputs. Considering this, our sandbox test cases allowing for a holistic analysis is a strong asset: The proposed tool for automatic test step generation includes optional integration testing to demonstrate and address the severity of the state-space explosion at this level, which nicely shows to be somewhat of a borderline of what can be implemented manually and what would require automatic generation.

### 6.2. Script Generator Design

The manually implemented test scripts and the requirements analysis provided by Bombardier defined Truth Tables 3–5. As discussed in the previous paragraph, we select setting all signals each step and offer optional integration testing to test shared outputs as first design goal. A challenge in implementing the script was the automatic test step selection. How can *which* test steps are relevant be determined? The actual challenge manifests in the step from having a textual description towards the actual implementation. How can it be ensured that the generated code matches the textual specification? What would be the *best* (i.e., clearest, most precise, shortest, impossible to misunderstand) contract

linking text to code? A second design goal is to manifest this connection. The third design request is the automatic output specification. Once the relevant inputs have been determined, the expected outputs shall be automatically added. Concluding, the three design goals are (i) integration testing, (ii) automatic test step identification, and (iii) automatic output calculation.

The code for the test script generator, which has been custom-tailored to this task, is available online (https://github.com/earthphoenix/BT, last visited 30 May 2022). The tool is implemented in Erlang. In order to execute it, set the relevant parameters in the header file `gen_script_config.hrl.` and compile the source code with `c(gen_script).` and execute it with `gen_script(start).` The script builds the required test cases according to the specification of the header file in five steps.

Step 0

Sets the desired options in the header file `gen_script_config.hrl`. The tool builds a file comprising (i) an initialization sequence, (ii) automatically generated code for each test step, and (iii) code shutting down the simulator or hardware. The initializing sequence is in file `tcHeader.txt`, the code for shutting down in file `tcFooter.txt`. These commonly change with different versions of the SuT. Once the desired outcome is specified in the header file, the following five steps generate the matching test steps:

  Step 1: Create Initial Vector
  Step 2: Make Boolean Combinations
  Step 3: Filter Relevant Steps
  Step 4: Add Expected Outputs
  Step 5: Replace with Code

Step 1

Creates a list with signal names depending on which test cases are activated in the header file. Notably, *shared* output signals are only to be added once. The output is a list containing unique atoms, one for each input and one for each output. Executing this step takes less than a second and is linear in complexity.

Step 2

Generates all permutations. With $4 + 8 + 4 = 16$ boolean input signals for test cases 1, 2, and 3, respectively, the resulting number of possible permutations for a full integration test is $2^{16} = 65,536$. In Erlang, this can be coded easily with pattern matching.

The first list $[A, B, \ldots]$ in Listing 5 contains placeholders, one for each input signal. The number of input signals comes from the previous step that selected the test cases to be included. The double pipe demarcates a list of variables, each allocated a list of all possible values (here Boolean). Generating all permutations commonly requires less than a second. This covers the first design goal and seamlessly allows for potential inclusion of non-binary values.

**Listing 5.** Implementing Permutations in Erlang.

```
[[A,B,..] A <- [true,false], B <- [true,false], ...];
```

Despite complexity being exponential, the result is just a list of lists containing Boolean signals. Its creation and storage at this point is not a challenge, even on limited hardware. The boundaries here have not been tested, but are expected to be sufficiently high to not be a problem, even when the number of test cases increases significantly.

Step 3

Filters those test steps that are actually deemed relevant. Since all permutations of input signals are generated anyway, the flag `?IMPORTANCE` in the header file overrides that filter when set to false. The filter is applied *before* the outputs are generated (step 4) to save some time by avoiding computing outputs of test steps that are filtered out regardless.

What makes a test step relevant? Formalizing this question seems simple. A test step is important if it reflects expected behavior mentioned in the textual requirements analysis.

The important test steps for the featured test cases for instance are those shown in the Truth-Tables 3–5.

For the manually generated test cases and steps, a test engineer looked at the textual description and then started coding. The *naïve* approach for the new tool would have been to simply write all desired input combinations mentioned in the textual description in an `if`-clause, acting like a bouncer in front of a club: `if` you are on the list you get in. Then, each possible allocation from step 2 can be checked for being on that list. The implementation is not much work for test cases comprising six or 13 relevant steps. Covering for a lot larger sets in the future on the other hand might yet be challenging. For example, consider a bouncer having to check *many* features of potential customers instead of six or 13. Therefore, on the one hand side, the number of entries and the size of the underlying KV-map is crucial in the first step. On the other hand side, integration testing becomes inherently more challenging due to the case distinction discussed in the next step, when the shared outputs are computed. Additionally, the second design goal described in Section 6.2 states *automatic test step identification*. Lastly, having this automatized might reduce the risk of missing and skipping relevant steps or adding irrelevant steps by mistake. After all, we can *formally* prove which test steps are included with an automatic method. The trick is to employ KV-maps for filtering, as discussed in Section 4. Figure 3 shows the mapping for TCs 1 and 3 in Figure 3a and for TC2 in Figure 3b.

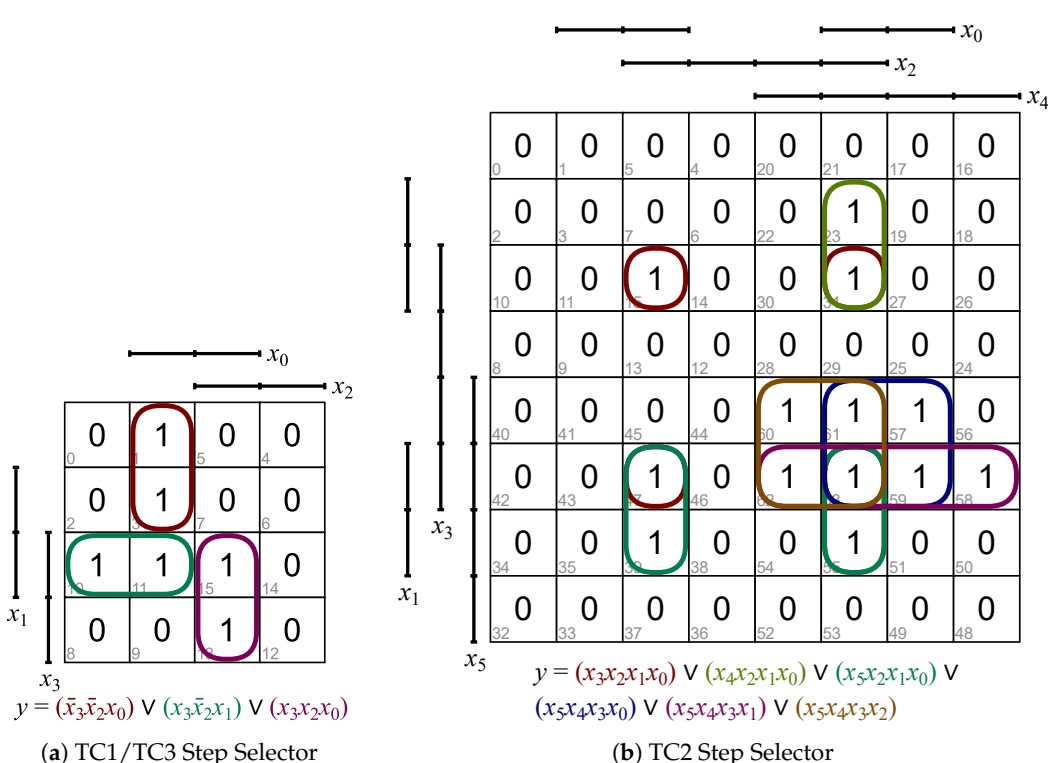

**(a)** TC1/TC3 Step Selector

$y = (\bar{x}_3\bar{x}_2x_0) \vee (x_3\bar{x}_2x_1) \vee (x_3x_2x_0)$

**(b)** TC2 Step Selector

$y = (x_3x_2x_1x_0) \vee (x_4x_2x_1x_0) \vee (x_5x_2x_1x_0) \vee$
$(x_5x_4x_3x_0) \vee (x_5x_4x_3x_1) \vee (x_5x_4x_3x_2)$

**Figure 3.** Selecting Relevant Steps with KV-Maps Figures were created with the online tool from https://www.mathematik.uni-marburg.de/~thormae/lectures/ti1/code/karnaughmap/, last visited 30 May 2022.

While the *input*-filter-maps of cases 1 and 3 are equal, their expected *output*-filter-maps—for which KV-maps (cf. for instance Table 2 on page 7) can be employed as well as a computation method—are different. The formulas underneath the graphs formalize the resulting DNFs.

In case of integration testing, a test case is considered to be relevant if all individual participatory input signals are relevant. For instance, consider a test step composing signals from test cases 1 and 2. If a signal-combination of test case 1 is considered relevant but a signal-combination of test case 2 is not, then the whole compound step is irrelevant.

The novel filtering based on first creating all permutations (in step 2) and then filtering them with KV-maps (in step 3) is expected to be less error-prone than the manual conversion. For once, the created KV-map can directly be compared to the textual requirements and each desired step can be appointed concretely to one 1 on the map (cf. Figure 3). Second, *all* signals are set each step so the sequence of steps can later be changed if desired, eradicating mistake propagation. The downside is that setting all signals consumes more time, which is not considered crucial since (i) it is only linear in complexity, (ii) setting a signal is considerably fast, and (iii) the testing process is parallelizable. The automatic code generation in the next step is fast and also not an issue. Finally, mistakes are expected to be observable much more easily, as a small mistake in the formal contract would result in a pattern of faults throughout the test steps that is easily observable.

These maps are the formal contract. It is possible to put a finger on a cell, linking it to both the textual requirements and the code line generated. If the test case was not generated but appears in the textual form (or vice versa), it shows up here.

Step 4

Generates the expected outputs for the relevant test steps. The challenge is that some test cases share some outputs. A shared output signal shall be **true** if all relevant test cases want it to be **true** and **false** otherwise. For instance, if test cases 1 and 2 do not raise an alarm but test case 3 does, the shared alarm lines shall still fire.

Step 3 provides a discussion—whether or not to utilize an automatic script for filtering the relevant test steps. Similarly, step 4 can discuss if the expected results shall be hard-coded or not. Since it might be desirable later to execute all permutations (i.e., by overriding the relevance filter, setting the `?IMPORTANCE` flag in the header file to **false**), hard-coding is not an option (or someone has to manually code the expected outputs for 65,536 steps). Again, the test engineer has to consult the requirements analysis and transfer *how* the output signals are expected to be set according to a given input.

The script iterates through the test cases. Starting with test case 1, a simple if-clause checks how all outputs are to be set. Test case two does the same, but distinguishes the cases of the shared signals being **true** or **false**: If the step is **true** in both test case 1 and 2, then the shared signal is **true**, otherwise it is **false** (analogously for test case 3). The challenge is that the position of the signals differs within the configuration depending on which test cases have been activated in the header file as discussed in step 1. This makes the case distinction inherently more challenging in the implementation.

Step 5

Replaces the generated atoms with text blocks. A step counter is added between each two configurations for tracing failed steps (i.e., false-positives or false-negatives). The generated code is then wrapped in the code from the header and footer files and written into an executable file.

### 6.3. The Scale of the TCMS

The scale of the presented industrial case study allows for a complete observation of the test suite composition with KV-maps. It does not matter if tests are failing or not, so large-scale studies focusing on classifying failing test cases, like the work by Jiang et al. [26], for instance, would not be suitable. The goal here is not to boast with size or to compete with efficiency, but to set up a small yet realistic proving ground for a proof-of-concept. The point is that KV-maps can aid the process as minimal formal contracts. If a mistake occurs and something was not tested as it should be, the KV-map is the ideal basis to find the cause. Mistakes are visible at an early stage as tests are generated by patterns and are thus less obfuscated.

Establishing KV-maps as a step between text and code has shown to be a valuable asset here. Common tools for requirement analysis like IBM DOORS allow for linking textual requirements to features. The KV-map serves the same purpose. Beyond that, it can even be applied to build the code directly based on it. While automatic test step generation from the IBM DOORS documentation is desired, it is only possible to link text fragments to

lines of code. There is no possibility to check if the implementation is semantically correct. While the same holds for the documentation towards the cells in a KV-map, the KV-map allows to check if the code is correct or even have it generated automatically.

Another benefit is that automatic code generation is much faster than manual implementation for test cases that are beyond sandbox size. Even for this small scale case study of three test cases, the manually implemented code targeted test cases only individually. An integration test was not required. However, if it would be required (in the sense that it would also be more complex than *if one fires, the joint signal shall fire*), it would have been too much code to write. Six test steps were written for test cases 1 and 3, and 13 for test case 2. The script builds $6 \times 13 \times 6 = 468$ test steps in about 3 s (Pentium i5-3317U@1.7 GHz). Setting `?IMPORTANCE` to **false** lets the system generate 65,536 steps taking 14 h on the same CPU. The vast majority of that time is spent writing the actual code to an executable file, not on its generation. The scaling is not linear, since the compound steps grow not only in the number of steps, but also in the number of signals that have to be set.

Exploiting KV-maps for automatic code generation is reasonable for tests above roughly 100 test steps (with a reasonably low number of variables). Despite the speed increase in code generation, the main benefit remains provably correct code.

## 7. Brief Discussion and Conclusions

The presented script is easily extendable beyond binary domains, as discussed in Step 2 in Section 6.2. While the complexity grows in the size of the domains that are to be permutated, further approaches like abstraction refinement, binning, or over-approximation can be exploited to cope with that. A greater challenge encountered when combining test cases is the case distinction regarding shared signals, however. Sorting a configuration based on which test case is activated in the header file has shown to be intricate.

The approach is easily applicable to other contexts than the TCMS. However, the TCMS here was not only the initial motivation to utilize KV-maps in testing, but also nicely allowed to demonstrate limitations and challenges of the method.

The customer's goal was to get relevant test steps automatically ("*magically*") from the textual description on "*the push of a red button*". That red button between requirements analysis and test script used to be a test engineer that had to conduct the manual implementation. This lead to the question in the introduction—how the process of test step implementation can be improved.

The traditional way of manually selecting test steps is prone to mistakes. Relevant steps can be forgotten or spelling mistakes might lead to wrong test sequences. Additionally, manual implementation commonly conservatively sets only those signals that change between two steps, allowing for mistake propagation. Although the *red button* generating required test steps for *any* textual requirements analysis is unrealistic, developing a tool that is custom-tailored to the provided use-case is not. KV-maps provide a powerful instrument in formalizing the informal filters from the textual requirements analysis before implementing them. They make up leeway for an automatized test generation.

The paper introduced a (closed source) TCMS as SuT. It discussed manually built test cases and pointed out challenges encountered like mistake propagation. As a solution, a script was presented exploiting KV-maps for filtering relevant test cases. Beyond that, KV-maps were utilized for specifying the expected output. The complexity of each part of the presented tool was discussed. The answer to the research question is: The process of test step implementation can be improved by KV-maps. General challenges and limitations have been addressed and the benefits have been pointed out.

As a next step, a model-based approach will be established for the same test scenario. This will complement the discussion, if in cases like these (i) the manual test generation, (ii) a custom-tailored script based on KV-maps or (iii) an *off-the-shelf* solution based on model-based test-step generation will be the *optimal* solution. The second option can also feature alternatives to KV-maps like the Quine–McCluskey [27] algorithm for comparison.

**Funding:** This research received no external funding.

**Data Availability Statement:** The script generating the test suite is available online, as stated above. The original requirements analysis is not available publicly, as discussed above.

**Acknowledgments:** Bombardier Transportation Sweden AB in Västerås, Sweden, supplied the use-case and access to the required facilities.

**Conflicts of Interest:** The author declares no conflict of interest.

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
