# Peer review of "Karnaugh-Veitch Maps as Minimal Formal Contract between Textual Requirements and Tests: A Use-Case Based Technical Analysis"

_electronics, doi:10.3390/electronics11152430_

Round 1

Reviewer 1 Report

Author presents a revised version of a manuscript that shows an approach to an automatic test generator based on Karnaugh-Veitch maps for a minimal formal contract between informal requirements documentation and implemented test steps.  

Despite the responses to my earlier comments point to the adaptation of text to correct some unclear Sections and problems, I regret to write that my main concerns have not been solved.  

The state of the art has been greatly improved with the addition of a new Section and this issue seems corrected. However, two very relevant issues are not solved: 

  1. The new method that the manuscript claims.

This is presented in Section 4 (The train control management System and KV-maps). However, it seems that the only novelty is the reduction in the number of Boolean combinations to test according to the simplification that KV miniterms offers. There is not a clear section with results that demonstrates the improvements associated to this method against the previous one.

This section is very similar to the section of previous version, except for second subsection that has been moved to Section Mistake propagation, after the State-of-the-art revision. 

  1. The methodology for assessing and the outcomes.

Author has added several paragraphs at the beginning of Testing section where he addresses the test cases used in the paper. However, despite this addition, the advantage of the new method is unclear. Where are the outcomes presented? How are they compared with outcomes of the previous technique?

The outcomes seem limited to the paragraph in lines 391 - 394.

The outcomes for the automatic test are not clear (Section “Script generator design”). I appreciate the effort and details exposed in it, but I cannot conclude a clear advantage as scientific contribution in this manuscript.  

In summary, despite I agree in the fact that KV maps may simplify and reduce the number of Boolean conditions to be tested, the current form of the manuscript does not clearly demonstrate these advantages either provide a clear method to apply it.

Author Response

Dear Reviewer,

thank you for your kind review. It helped me improve the article.

  1. You write “There is not a clear section with results that demonstrates the improvements associated to this method against the previous one.”
    1. The abstract has been rewritten to tackle that concern. I understand now, how this was unclear in the previous version. The thought is:
      1. KVmaps allow to display requirements in a computer-editable way
      2. They further allow to specify which test steps are relevant (i.e. which rows of a complete value table are to be checked)
  • With both requirements and “relevances” at hand, you can automatically generate the steps.
  1. Line 76 now also contains a sentence in that regard: “The advantage of KV-maps the test engineer only has to concern with few Boolean formulas
    and their relation to textual requirements instead of confusing test steps of a test case and their relation.”
  2. Line 20/21 already states in the introduction “While implementing
    test steps manually is inherently prone to mistakes, their detection is difficult as the test result does not state if the test itself is correct (i.e., as intended)”
  3. Section 2 is dedicated to raise and answer the questions “Why are the test cases implemented manually in the first place? What are the alternatives?”
  4. The whole TCMS is dedicated as example to demonstrate how applying KV-maps in testing can improve manual testing as state-of-the-art.
  5. Line 368 points out the mistake detected by the method on the example that was found by utilizing KV-maps.
  6. The paragraph starting in line 377 discusses the advantages and disadvantages of setting all signals each step as trade-off between additional time against exchangeable step sequence and avoiding mistake propagation.
  7. Step 3 in line 454 discusses the boundaries of the manual approach and even, what this implies if the same approach would be followed for automatic test step generation instead of KV-maps.
  8. Finally, the paragraph starting in line 573 concludes the whole argument.

So, from my point of view the article discusses this very important question during the whole proceeding, raising the question in the beginning, pointing out benefits of various aspects of the method, and finally also concludes them. This remark pointed out an important point in the first round and I hope to have improved the article in that regard.

  1. You write “However, despite this addition, the advantage of the new method is unclear.” While this point sound very much like the first point you raise, I figure it does not only concern the comparison of manual implementation vs. the proposed method. Line 76 states that already before the actual implementation, KV-maps are an advantage as they provide a minimal formal contract, linking the textual requirements documentation to code, thus overcoming the state-of-the-art.

I am aware that KV-maps are no novel technique and well-established in circuitry-design. Nevertheless, a whole department of developers and test-engineers did not apply KV-maps for the TCMS. The article shall be published as reference for future cases when KV-maps can be applied.

Kind regards,

Nils Müllner

Reviewer 2 Report

The manuscript still suffers from ambiguity, and it is not well represented. The authors still suddenly use terms assuming the reader knows everything. The authors need to look at the manuscript in its big picture:

·       does the introduction sufficiently clarify the scope of the manuscript? Partially. The ideas were not connected, approach was not introduced properly. Language is much weak and needs major editing. Even the authors’ responses include mistakes!!

·       Connections used are wide and confusing. Some information were provided in the intro that they don’t really belong here! Technical details should be discussed in a separate section (e.g. methods). I recommend the authors to focus more on the subject. Make each paragraph discuss one or two ideas and only present major ideas in the intro. Leave details for later.

·       Does the Related Works section represent and discuss sufficient material? Not really. This section discusses few works with the aim of clarifying the methods used in the paper in hand, which is not usually the aim of this section. This section should be an unbiased area in which the authors only discuss methods and results of other works. The novelty of the proposal in hand must be declared in the intro or after the results in a Discussion section. Referring to my First Major Comment in the previous review round, this version did not provide a ‘good’ Related Works section and did not address the outperformance requirement. The proposal must be tested against other methods to show it is a good solution, otherwise how can the reader know? Benchmarking and utilizing a suitable performance approach is a major task researchers usually handle. Note that even as a Proof-of-Concept, the authors need to provide some results so that other researchers can compare with or even future versions of the proposed solution can be shown better than previous versions. Refer to methods that has the same objectives of yours with different approaches. Show that your approach is better somehow!

·       Researchers usually address the comments in the response and in the manuscript with exact details of the modifications performed. Responding to a comment by ‘adapted’, or ‘done’ only is somewhat disrespectful and not helpful for a quick process. For example, I had a comment regarding section 2.2 in the previous section. Now section 2 became 4 (the authors did not mention that!) and there is no section 4.2 anymore to check!

·       Tables and Figures and their captions are still of much poor quality   

Author Response

Dear Reviewer,

thank you for your kind review. It helped me improve the article.

  1. You write “The ideas were not connected, approach was not introduced properly. Language is much weak and needs major editing. Even the authors’ responses include mistakes!!”. The first point is that the ideas were not connected. I hope to satisfy that concern with the new version. The “red thread” should now be easier to follow. The second point regards the language quality. The article has been proofread and edited. Please let me know concrete mistakes to correct if there are any left.
  2. You write “Connections used are wide and confusing. Some information were provided in the intro that they don’t really belong here! Technical details should be discussed in a separate section (e.g. methods). I recommend the authors to focus more on the subject. Make each paragraph discuss one or two ideas and only present major ideas in the intro. Leave details for later.” The introduction now contains only relevant information and details are provided where needed. In case concrete adaptations are required, please be so kind to state those concrete improvements.
  3. You write “Does the Related Works section represent and discuss sufficient material? Not really. This section discusses few works with the aim of clarifying the methods used in the paper in hand, which is not usually the aim of this section. This section should be an unbiased area in which the authors only discuss methods and results of other works. The novelty of the proposal in hand must be declared in the intro or after the results in a Discussion section. Referring to my First Major Comment in the previous review round, this version did not provide a ‘good’ Related Works section and did not address the outperformance requirement. The proposal must be tested against other methods to show it is a good solution, otherwise how can the reader know? Benchmarking and utilizing a suitable performance approach is a major task researchers usually handle. Note that even as a Proof-of-Concept, the authors need to provide some results so that other researchers can compare with or even future versions of the proposed solution can be shown better than previous versions. Refer to methods that has the same objectives of yours with different approaches. Show that your approach is better somehow!” The points you raise are important. As far as I understand there are two issues: The related work section requires more references and the benefits of the presented method over other methods is missing.
    1. KV are used here as a language for formalizing test specification and input for test generation. This is appropriate for the problem area and corresponds to other methods, which are also based on logic or sequential models (automata). The issue with related work is that KV-maps are a well-established method from circuitry design. They are very basic knowledge that are commonly taught in the first or second year in computer science. Yet, a whole department of developers and test-engineers was not aware they can be applied to the use-case, the TCMS. Their application to testing is new. There is no literature about KV-maps in testing similar systems. This is not state-of-the-art. There is only literature on KV-maps and then there is literature on testing (similar systems). The literature on testing inherently cannot be exhaustive in an article, but the selected related work discusses those papers that allow to provide a good background to the reader. If there are concrete references that are important to the paper, please let me know. The paper clearly states the improvements over the state-of-the-art.
    2. The second issue is the improvement that comes with utilizing KV-maps. In terms of benchmarking, it is difficult to formalize a metric. The number of mistakes found by KV-maps compared to manual implementation is all, which means one mistake was found for the TCMS and the method proved there is no other mistake. The time required for coding depends very much on the developer and is not a subjective metric. The methodological improvements regarding integration testing are widely discussed.
      1. The abstract has been rewritten to tackle that concern. I understand now, how this was unclear in the previous version. The thought is:
        1. KVmaps allow to display requirements in a computer-editable way
        2. They further allow to specify which test steps are relevant (i.e. which rows of a complete value table are to be checked)
        3. With both requirements and “relevances” at hand, you can automatically generate the steps.
      2. Line 76 now also contains a sentence in that regard: “The advantage of KV-maps the test engineer only has to concern with few Boolean formulas
        and their relation to textual requirements instead of confusing test steps of a test case and their relation.”
  • Line 20/21 states in the introduction “While implementing
    test steps manually is inherently prone to mistakes, their detection is difficult as the test result does not state if the test itself is correct (i.e., as intended)”
  1. Section 2 is dedicated to raise and answer the questions “Why are the test cases implemented manually in the first place? What are the alternatives?”
  2. The whole TCMS is dedicated as example to demonstrate how applying KV-maps in testing can improve manual testing as state-of-the-art.
  3. Line 368 points out the mistake detected by the method on the example that was found by utilizing KV-maps.
  • The paragraph starting in line 377 discusses the advantages and disadvantages of setting all signals each step as trade-off between additional time against exchangeable step sequence and avoiding mistake propagation.
  • Step 3 in line 454 discusses the boundaries of the manual approach and even, what this implies if the same approach would be followed for automatic test step generation instead of KV-maps.
  1. Finally, the paragraph starting in line 573 concludes the whole argument.

The outlook in lines 588ff states that the same SuT can be further compared to off-the-shelf solutions in another paper. This paper is confined in contributing only KV-maps as an asset in testing on the example of the TCMS.

  1. You write “Researchers usually address the comments in the response and in the manuscript with exact details of the modifications performed. Responding to a comment by ‘adapted’, or ‘done’ only is somewhat disrespectful and not helpful for a quick process. For example, I had a comment regarding section 2.2 in the previous section. Now section 2 became 4 (the authors did not mention that!) and there is no section 4.2 anymore to check!” I am very sorry and did not mean to be disrespectful. Due to the constructive feedback the article was changed in several rounds so the changes became hard to track.
  2. You write “Tables and Figures and their captions are still of much poor quality”. The captions are now improved. The quality of the figures and tables should be optimal as they are scalable vector graphics and/or text.

As far as I understand your main concerns, the major shortcomings of the article are i) insufficient related work and ii) lack of comparison to other solutions. I hope I was able to address these concerns above. Relevant related work is addressed in the article and if there is a certain publication that is relevant, I am simply unaware of it. In that case, please be so kind to point it out to me. The lack of comparison to alternative solutions is based on the fact that the one alternative solution, the state-of-the-art, is simply manual implementation. To this, the proposed method is compared in every aspect. Yet, concrete metrics for benchmarking are redundant here.

I hope I have addressed all the issues raised sufficiently.

Kind regards,

Nils Müllner

Reviewer 3 Report

My comments have been, more or less, addressed. Please correct the typos below and add some general references when Domain-Specific Languages (DSLs) are mentioned for the first time.

Typos:

orinigal focus  -> original focus

The contributions of this paper are -> The contributions of this paper are:

 domain specific language -> domain-specific language

Author Response

Dear Reviewer,

thank you for your kind review. The concerns raised have been corrected. It helped me improve the article.

Kind regards,

Nils Müllner

Reviewer 4 Report

Paper is written clearly and corectly. The research is covered and supported by theoretical part and experimental part. The results are presented and they are clearly described.

Paper needs only formal corrections:

- avoid you-form (e.g. p. 14)

- consider unified writing for numerical calculations, e.g. for multiplication \cdot and * are used (6 . 13 . 6 and 6 * 13 * 6)

Author Response

(The authors gave the same response as above.)

Round 2

Reviewer 1 Report

After reading again the author responses and manuscript revised version, my perception from the scientific contribution of the manuscript (related to outcomes and evaluation methodology) is similar. However, I am aware that the author know-how concerning engineering tests in TCMS shows that this method could be applied with success in such a department. Certainly, an interesting technical - scientific contribution could be achieved if KV – maps were applied after manual tests during a real industrial testing. A subsequent comparison of the obtained outcomes, including proper defined metrics, could help to quantify the true benefits of the proposed KV maps in testing.  

I may suggest several changes that could clarify the scope of the present manuscript under the present form:

  1. Title. It could be completed with something similar to“: a use case based technical analysis”
  2. Abstract. It could be slighted modified.

Line 9- - : Change as (or similar): “The approach is APPLIED on a real-world industrial use-case, a train control management system”

Line 11 - : Change as (or similar) “The method MAY HELP to disclose flaws in CURRENT manual implemented tests, as this preliminary analysis suggests”

  1. Introduction. It could be slighted modified.

Line 95- : Remove “Section 7 comparatively shows benefits and drawbacks of both manual and automatic approaches” (since that section does not address that comparison).

  1. Challenge and Prospect Section (7) could be moved properly to Conclusion section, which in turn should be renamed as “Brief discussion and conclusion”.

Author Response

Dear Reviewer 1,

I reply to your second review that I copied to the end of this letter.

To process the use case in different industrial tools in comparison with the demonstrated custom-tailored KV-maps approach will result in hopefully another interesting article. You found important parts that I hopefully improved to your satisfaction.

Kind regards,

Nils Müllner

Reviewer 2 Report

Major comments:

·       Line 89: “3. showing how KV-maps can be further exploited for an automatic (and provably correct) test generation.” This is somewhat inaccurate! If a research is proposing something he has to show how it works or how it can be used/exploited! Otherwise the method is rejected. This is not a contribution but a way to show the novelty and usability of the proposal, which is mandatory in scientific research. Contributions are the things that are additions to the body of knowledge, not the things a researcher have done in order to contribute an addition to the body of knowledge.

·       The authors claim that it is difficult to formalize an evaluation metric, and that there are no literature about KV-maps in testing similar systems. However, the claimed novel contribution of the paper is using KV-maps, which means that the authors need to compare this novel method with other methods. Meaning that I’m not asking the authors to find a previous work that uses KV-maps to generate test-suits but I’m asking them to find previous works that simply generate test-suits using any method and compare with these. Searching google scholar for “automatic test generation” in the title returned thousands of articles. Only in 2022 there have been 25 published papers. The authors requested to provide relevant literature, which is basically not the job of the reviewer to study the literature and highlight the proposal’s novelty, but the authors job ! Nevertheless, examples include, but not limited to, Genetic algorithms have been used in [1], ASMETA was used in [2], reliable comparisons were presented in [3]. In these works, benchmarks such as code coverage, number of iterations, branch coverage, understandability scoring, robustness of test cases, etc. Authors can refer to what ever they want in the literature, specifically methods proposed recently, and find common benchmarks to compare with the proposal experimentally. Then the novel contributions can be good enough to be examined by reviewers!

·       I still have several comments regarding the presentation of the manuscript and the connection between the ideas. The authors requested this reviewer to state line-by-line or sentence-by-sentence, each mistake or ambiguity of the manuscript. I believe, this is not a task for a reviewer but a general comment regarding the recommendation for professional editing or enhancing the connection between ideas, should be enough. Several minor misrepresentations made the manuscript look as a class assignment rather than a research paper. I will provide some examples in the minor comments section. But those are not the only points to consider in future revisions.

·       The ‘Related Works’ section, again, should not be dedicated to discussing research questions and the novelty of the proposal, or justifying why it is not inclusive, or setting research questions and directing the reader where relevant information are discussed later, nor to provide definitions and explanations of the testbed! This section must simply, and only, discuss the most relevant works in the literature and the limitations of these works. Highlight the limitations that are addressed by the proposed method and later prove its outperformance with experimental results. I mentioned this as a major comment previously and I didn’t find it to be addressed in the new version of the manuscript.

·       I have another major comment regarding experimental testing and evaluation against other relevant works and this comment has not been addressed in the new version.

Minor comments (examples):

·       Line 17: Agile is a very widely used term for different purposes. Define it before using it.

·       Line 24: This paper aims at improving the trade-off between the costs of testing and the potential risk of having overlooked a potential safety hazard.

·       Since the TCMS technical details (i.e. code and technical description of workflow )

·       Reference needed for IBM DOORS at line 75

·       What I meant by “technical details should not be in the introduction” is the description provided between lines 57—84. Note that you can keep these info but then this means that the proposed method is only for this and not generic. Accordingly, the title of the paper should be changed followed by a modification to the purposes. Otherwise, omit such info and put them in their right place. You can mention this TCMS application however in one or two sentences as only an example of where the proposed method can be used.

·       introducing the TCMS and its test cases as industrial case study is NOT a contribution. The Authors are not proposing TCMS and using it as test case for the proposed method is one way that may, or may not, be good enough to highlight the novelty and contribution of the proposed method.

·       Line 261: put each table and figure on separate lines

·       Table captions on top and figure captions on bottom. Centralize the captions.

·       Line 365: give some space for the listing

·       Line 526: is this a heading or a sentence? Make it look as it is intended to be

·       Section 7 is very limited, either extend or omit and move text to other relevant sections.

Language weaknesses(examples):

·       Line 16: the test>> test-case generation for whether or testing whether…

·       Line 69: This motivated to reason how..!

·       Line 109: ‘The’ should not be put alone on a single line!

·       Line 221: ‘Example’ should not be put alone on a single line!

References:

[1]: Liu, Zhenpeng, et al. "Automatic Generation of Test Cases Based on Genetic Algorithm and RBF Neural Network." Mobile Information Systems 2022 (2022).

[2]: Bombarda, Andrea, Silvia Bonfanti, and Angelo Gargantini. "Automatic Test Generation with ASMETA for the Mechanical Ventilator Milano Controller." IFIP International Conference on Testing Software and Systems. Springer, Cham, 2022.

[3]: Alshammari, Majdah, Mohammad A. Mezher, and Khaled Al-utaibi. "Automatic Test Data Generation Using Genetic Algorithm for Python Programs." 2022 2nd International Conference on Computing and Information Technology (ICCIT). IEEE, 2022.

Author Response

%% colorized test in the appanded word file

Dear Reviewer 2,

thank you for your comments. I have to admit that I perceive the role of author and reviewer a bit differently. From a reviewer I hope to get constructive feedback that helps me, the author, to improve the article, or, in the worst case, sufficient evidence that the article does not qualify for publication. The feedback I received from you (and please correct me if I misunderstand you here) falls mainly into three categories:

  1. Lack of references/citations
  2. Lack of comparison with competing solutions
  3. Weak language

Along with the first point, you provide statistics from a Google Scholar search. Of course, I searched for relevant literature myself. Yet, due to the nature of the article (applying a method from the 50ies on a modern problem that has not been done before) there is not much literature on the topic. Selected references tackle adjacent research fields like genetic algorithms. Yet, your expectations are i) not clear and ii) unrealistic. This paper is not a survey paper aiming at citing all relevant literature. The sources you provide are tackled in my response in detail. In my opinion, they all disqualify to be added to the bibliography. They also fail to show that the part on related work is insufficient.

The second point is comparing the featured method to competing alternatives. Utilizing KV-maps for testing the TCMS is a method that is adequate here. Establishing metrics for comparison to other methods is a different topic (one, that will hopefully be tackled in a follow-up paper). This is clearly stated in the conclusion.

The third point is weak language. Again, I fail to see this point, especially if you provide no concrete examples. The examples provided in the minor comments do not seem so strong themselves. Again, please find the individual responses to each remark in the following part.

Concluding, I fail to see constructive feedback to improve the quality of the article or a substantial point disqualifying the article. Already in the previous round I asked for concrete points to improve the article. I explained the scope of the article multiple times (i.e. that the article is neither a survey that needs more references, nor that it is an exhaustive benchmark). Yet, the arguments return.

As an author it is challenging to find the literature a reviewer wants to see when even the reviewer does not know that literature (“not the job of the reviewer”). If you personally find the language weak, then please show me those weaknesses (“I believe, this is not a task for a reviewer”). The journal has been through multiple reviews and all concerns raised so far have been responded to. The weaknesses you point out are controversial.

Yet, I value your feedback, I realize that you put a lot of thought and work into it, and have taken it seriously. I adapted several points (please find the details in the following part) and hope i) to have brought the article to a quality that you deem sufficient now, and ii) that I could provide sufficient argument that the article is not a benchmark nor a survey. It simply shows how KV-maps can be exploited for test-cases such as the TCMS. I agree that to some degree it might seem trivial. Yet, when a whole department working for years on such a system still requires to hire a person to implement the test steps by hand, I think a publication can help to provide a reference for the future, showing how KV-maps are a valuable asset for such cases.

Thank you again for your review and the helpful comments. I hope to have now improved the article to meet your expectations.

Kind regards,

Nils Müllner

Major Points

  • Line 89: “3. showing how KV-maps can be further exploited for an automatic (and provably correct) test generation. This is somewhat inaccurate! If a research is proposing something he has to show how it works or how it can be used/exploited! Otherwise the method is rejected. This is not a contribution but a way to show the novelty and usability of the proposal, which is mandatory in scientific research. Contributions are the things that are additions to the body of knowledge, not the things a researcher have done in order to contribute an addition to the body of knowledge.

èLine 89 è: with KV-maps allowing to define which test steps are relevant, showing how KV- maps — specifying both requirements and relevance — can be further exploited for an automatic (and provably correct) test step generation. I fail to see the inaccuracy here, but I tried to write it more clearly and more consistent with the abstract. The present paper shows en detail how KV-maps are to be utilized to further exploit them for automatic test step generation. The point here is not to demonstrate that the method works, but to discuss how it works on a concrete example. This should also be clear since the intricacies of the whole approach, like for instance the order of joint signal lines in combinatorial testing, are exhaustively discussed.         

  • The authors claim that it is difficult to formalize an evaluation metric, and that there is no literature about KV-maps in testing similar systems. However, the claimed novel contribution of the paper is using KV-maps, which means that the authors need to compare this novel method with other methods. Meaning that I’m not asking the authors to find a previous work that uses KV-maps to generate test-suits but I’m asking them to find previous works that simply generate test-suits using any method and compare with these. Searching google scholar for “automatic test generation” in the title returned thousands of articles. Only in 2022 there have been 25 published papers. The authors requested to provide relevant literature, which is basically not the job of the reviewer to study the literature and highlight the proposal’s novelty, but the authors job! Nevertheless, examples include, but not limited to, Genetic algorithms have been used in [1], ASMETA was used in [2], reliable comparisons were presented in [3]. In these works, benchmarks such as code coverage, number of iterations, branch coverage, understandability scoring, robustness of test cases, etc. Authors can refer to whatever they want in the literature, specifically methods proposed recently, and find common benchmarks to compare with the proposal experimentally. Then the novel contributions can be good enough to be examined by reviewers!

èHere, KV-maps are used to formalize the spec and to derive test cases/steps. There are certainly other methods to formalize and generate test steps. But KV-maps are adequate to the domain here with the Boolean I/O. Test generation for code coverage is not an issue here as we aim for spec coverage. (Virtually) all more advanced test generation methods assume a formal input: Formal specs (ASM or other models) or code itself. Here, KV-maps take the role of the formal model

Suits for test generation are abundant. Literature on the topic is abundant. There are survey-papers about survey-papers aiming to tackle just a fraction of the topic (I am currently a co-author in one of those, focusing just on model-based testing). The blunt selection of “any method” seems somewhat unscientifical. Especially since the exploitation of KV-maps was a result of getting the TCMS as SuT in the first place. The method KV-maps are compared to is the manual implementation, as this is the state-of-the-art. I concur that a study of related work is not the job of the reviewer. Related work that was deemed relevant was included in the paper. Regarding the selected publications by the reviewer (thank you for bringing this up for discussion):

[1] regarding genetic algorithms, Alrawashdeh2020 is already cited and discussed exemplarily (line 109). Utilizing genetic algorithms is helpful when test coverage is to optimized (which is a completely different topic). Similarly, Tomita2019ASM is cited, utilizing with Monte Carlo a different strategy. This goes more into the direction of search space exploration. One of my favorite papers in that domain is Ellen2015 [4], which I do not discuss on purpose in the present paper, since search-based techniques are not directly relevant to show the benefits of utilizing KV-maps.

[2] The ASMETA paper follows the four steps “1. Generation of abstract tests from a ASMETA model; 2. Optimization of the abstract tests; 3. Concretization of the abstract tests in GoogleTest; 4. Execution of the concrete tests on code” (which is similar to the steps laid out in Section 6.2 of my paper). Yet, the precondition to acquire tests from abstract state machine specifications provides a completely unrelated use-case. The TCMS is not an abstract machine, but a real implemented system that is not available as ASM. The customer required implemented test steps and the state-of-the-art is to build them by hand. Remodeling the whole system (not just the test cases) as ASM lifts the complexity of generating test cases to a whole new level, which is reason enough to not dive into this unrelated discussion in the paper.

[3] Is yet another paper regarding genetic algorithms. The same argumentation as for [1] applies. Again: Fitness and coverage are irrelevant to the TCMS, and hence is literature from that domain. Beyond that, “reliable” is a term for instance defined by Trivedi [5] who also won an award (IEEE reliability-society-lifetime-achievement-award) for this [6]. It is here used in the wrong context. Beyond that, the authors write only in their future work section that “From the extensive deployment of the proposed mechanism, it is proven that further enhancements and
reliability can be achieved by developing additional procedures in different deployment aspects.” As far as I understand, the authors refer to the coverage, which is not the scope of the present paper.

I am well aware that the software testing community is very strong and novel techniques must prove themselves before being considered worthy. For that reason, it is important to provide a novel method with some kind of benchmark that clearly show that the new method outperforms its competitors.

 Yet, the contribution of the present paper is not a tool, but merely a strategy to incorporate KV-maps to avoid mistakes in the process. This is something completely different to genetic algorithms in testing or utilizing the system model in form of an ASM to generate test cases. The benefit of the proposed method lies in its verifiably correctness. As competitor, the manual method was selected and it was shown, how KV-maps help in discovering mistakes. With KV-maps exploited for both requirements and relevance, automatic test step generation becomes possible. The focus here is to (formally specify and) generate all those test steps that are relevant. Other research directions focusing on genetic algorithms, optimal control, the perfect balance between exploitation and exploration of the search space, … are simply different research topics. Nevertheless, some selected publications are presented in that regard to demarcate the present paper from them. With the abundance of publications, I hope it is clear, that an exhaustive literature review is impossible and also not the scope of the present paper.

  • I still have several comments regarding the presentation of the manuscript and the connection between the ideas. The authors requested this reviewer to state line-by-line or sentence-by-sentence, each mistake or ambiguity of the manuscript. I believe, this is not a task for a reviewer but a general comment regarding the recommendation for professional editing or enhancing the connection between ideas, should be enough. Several minor misrepresentations made the manuscript look as a class assignment rather than a research paper. I will provide some examples in the minor comments section. But those are not the only points to consider in future revisions.

èIt is challenging to address a general remark such as this one. The research paper has been reviewed multiple times by many peers and those flaws that have been brought to attention have been fixed. It is intrinsically impossible to fix a flaw you do not see or understand. Hence, it is required to state what has to be improved as concretely as possible. I am very thankful for the flaws you discovered and list later.

  • The ‘Related Works’ section, again, should not be dedicated to discussing research questions and the novelty of the proposal, or justifying why it is not inclusive, or setting research questions and directing the reader where relevant information are discussed later, nor to provide definitions and explanations of the testbed! This section must simply, and only, discuss the most relevant works in the literature and the limitations of these works. Highlight the limitations that are addressed by the proposed method and later prove its outperformance with experimental results. I mentioned this as a major comment previously and I didn’t find it to be addressed in the new version of the manuscript.

èThe related work section discusses selected publications to show the reader relevant context. This includes a demarcation of the paper’s contribution the contributions of other papers. The discussion of the contributions of the present paper is not limited to the related work section, but only referred to, where necessary in the context of related contributions. Excluding these references would leave the reader alone to figure out themselves, how the contributions of the present paper fit into the state-of-the-art.

  • I have another major comment regarding experimental testing and evaluation against other relevant works and this comment has not been addressed in the new version.

èThe terms “experimental testing” and “evaluation” did not occur in your previous review. I answered each point directly. As far as I understand we have a different view of what the contribution of the present paper is. The goal of the paper is to show on the TCMS how KV-maps can be utilized, and to show how this improves the generation of test steps regarding mistake prevention and automation of the process. The goal is not to survey options for modeling a system or requirements, nor a survey on search-based methods. The contribution to the state-of-the-art has been presented several times now (in the paper itself and in the review-responses). It is also addressed in the conclusion (line 589) that a comparison to off-the-shelf solutions can be future work (i.e. a different contribution).

Minor comments (examples):

  • Line 17: Agile is a very widely used term for different purposes. Define it before using it.
  • On the contrary, I find Agile Software Development to be quite a well coined term (see also https://en.wikipedia.org/wiki/Agile_software_development). Defining the term in the scope of the paper is not required here and would not contribute to the quality of the paper.
  • Line 24: This paper aims at improving the trade-off between the costs of testing and the potential risk of having overlooked a potential safety hazard.
  • That is a quote from the paper. I fail to see what should be wrong about it.
  • Since the TCMS technical details (i.e. code and technical description of workflow )
  • Again, I am uncertain what this comment refers to. The code is provided along with documentation. The workflow is described in the paper.
  • Reference needed for IBM DOORS at line 75
  • Reference for IBM Doors is already provided earlier in line 59. Citing a reference at each occurrence of DOORS is redundant.
  • What I meant by “technical details should not be in the introduction” is the description provided between lines 57—84. Note that you can keep this info but then this means that the proposed method is only for this and not generic. Accordingly, the title of the paper should be changed followed by a modification to the purposes. Otherwise, omit such info and put them in their right place. You can mention this TCMS application however in one or two sentences as only an example of where the proposed method can be used.
  • I get the point and hope to have tackled it by having changed the title. Although the method is applicable in permutation testing in general, the paper is inherently confined to demonstrate it on the featured TCMS. The benefit of featuring some technical detail in the introduction (i.e. detail that provides a first glimpse to the reader of what follows later) in my opinion outweighs the disadvantages. Certainly, providing such information shall not limit the scope of the paper. Yet, I can see how one might perceive it as “too much” and change the title of the paper accordingly.
  • introducing the TCMS and its test cases as industrial case study is NOT a contribution. The Authors are not proposing TCMS and using it as test case for the proposed method is one way that may, or may not, be good enough to highlight the novelty and contribution of the proposed method.
  • That is true. Formalizing for instance the requirements the relevance of the steps exemplarily for the TCMS are contributions.
  • Line 261: put each table and figure on separate lines

è Although I fail to see the point, the table and the figure are now in separate lines.

  • Table captions on top and figure captions on bottom. Centralize the captions.
  • I use here the style file provided by the journal. If the editor prefers to change this, I will adapt that style file accordingly.
  • Line 365: give some space for the listing
  • The space to the right increased (i.e. right border of the box moved to the right)
  • Line 526: is this a heading or a sentence? Make it look as it is intended to be
  • It was a paragraph, now it is changed to be a subsection
  • Section 7 is very limited, either extend or omit and move text to other relevant sections.
  • Sections 7 and 8 have been joined.

Language weaknesses(examples):

  • Line 16: the test>> test-case generation for whether or testing whether…
  • Changed, although the language was not incorrect nor was readability or quality of language increased.
  • Line 69: This motivated to reason how..!
  • Section~\ref{sec:Mistake Propagation} shows how manual implementation caused undetected mistakes in the test routine, which motivated to reason how such mistakes can be avoided.
  • Line 109: ‘The’ should not be put alone on a single line!
  • In the current version there is no “The” on a single line in line 109.
  • Line 221: ‘Example’ should not be put alone on a single line!
  • This is the heading of a paragraph

References:

[1]: Liu, Zhenpeng, et al. "Automatic Generation of Test Cases Based on Genetic Algorithm and RBF Neural Network." Mobile Information Systems 2022 (2022).

[2]: Bombarda, Andrea, Silvia Bonfanti, and Angelo Gargantini. "Automatic Test Generation with ASMETA for the Mechanical Ventilator Milano Controller." IFIP International Conference on Testing Software and Systems. Springer, Cham, 2022.

[3]: Alshammari, Majdah, Mohammad A. Mezher, and Khaled Al-utaibi. "Automatic Test Data Generation Using Genetic Algorithm for Python Programs." 2022 2nd International Conference on Computing and Information Technology (ICCIT). IEEE, 2022.

[4] https://www.semanticscholar.org/paper/Statistical-model-checking-for-stochastic-hybrid-Ellen-Gerwinn/34fae206b59fdff3a15feeb2c3fac2a517cb2895

[5] https://www.wiley.com/en-in/Probability+and+Statistics+with+Reliability%2C+Queuing%2C+and+Computer+Science+Applications%2C+2nd+Edition-p-9780471333418

[6] https://ece.duke.edu/about/news/trivedi-wins-ieee-reliability-society-lifetime-achievement-award

Round 3

Reviewer 2 Report

 Thank you for addressing most of my comments.  The authors state in their last revision that they are considering the comparison of the proposed methods, with related works, in future papers. This makes the presented contributions of the paper incomplete, which makes the paper a Position paper rather than a Journal Article.

Author Response

Dear Reviewer 2,

Thank you for your quick reply. Relevant selected publications are discussed sufficiently within the present article. Modeling the use-case in other tools is inherently a different topic than establishing the exploitation of KV-maps for requirements and relevance as an adequate method for the TCMS in the scope of generating test steps.

The present article is – in my opinion - clearly not a position paper, where one argues the pros and cons of something, but instead provides a concrete contribution to the state-of-the-art, along with a detailed example in a scientific neutral way without any positioning.

Providing a future outlook of what further contributions can be achieved does not mean that these future contributions must already be part of the present research. Instead, it demarcates the actual contribution from what was (related work) and what will be (future work).

Kind regards,

Nils Müllner
